# Exosomes Derived from Fisetin-Treated Keratinocytes Mediate Hair Growth Promotion

**DOI:** 10.3390/nu13062087

**Published:** 2021-06-18

**Authors:** Mizuki Ogawa, Miyako Udono, Kiichiro Teruya, Norihisa Uehara, Yoshinori Katakura

**Affiliations:** 1Graduate School of Bioresources and Bioenvironmental Sciences, Kyushu University, Fukuoka 819-0395, Japan; ogawa.mizuki.908@s.kyushu-u.ac.jp (M.O.); kteruya@grt.kyushu-u.ac.jp (K.T.); 2Faculty of Agriculture, Kyushu University, Fukuoka 819-0395, Japan; mudono@grt.kyushu-u.ac.jp; 3Department of Molecular Cell Biology and Oral Anatomy, Faculty of Dental Science, Kyushu University, Fukuoka 812-8582, Japan; ueharan@dent.kyushu-u.ac.jp

**Keywords:** telomerase reverse transcriptase, keratinocyte–hair follicle stem cell interaction, exosomes, telogen–anagen transition, hair cycle regulation

## Abstract

Enhanced telomerase reverse transcriptase (TERT) levels in dermal keratinocytes can serve as a novel target for hair growth promotion. Previously, we identified fisetin using a system for screening food components that can activate the TERT promoter in HaCaT cells (keratinocytes). In the present study, we aimed to clarify the molecular basis of fisetin-induced hair growth promotion in mice. To this end, the dorsal skin of mice was treated with fisetin, and hair growth was evaluated 12 days after treatment. Histochemical analyses of fisetin-treated skin samples and HaCaT cells were performed to observe the effects of fisetin. The results showed that fisetin activated HaCaT cells by regulating the expression of various genes related to epidermogenesis, cell proliferation, hair follicle regulation, and hair cycle regulation. In addition, fisetin induced the secretion of exosomes from HaCaT cells, which activated β-catenin and mitochondria in hair follicle stem cells (HFSCs) and induced their proliferation. Moreover, these results revealed the existence of exosomes as the molecular basis of keratinocyte-HFSC interaction and showed that fisetin, along with its effects on keratinocytes, caused exosome secretion, thereby activating HFSCs. This is the first study to show that keratinocyte-derived exosomes can activate HFSCs and consequently induce hair growth.

## 1. Introduction

Telomerase reverse transcriptase (TERT) maintains telomere length and is known to be active in stem cells as well as cancer cells. Sarin et al. (2005) reported that when TERT is transgenically introduced into mouse skin, the hair follicle cycle shifts from the telogen phase, the quiescent phase, to the anagen phase, the active phase, resulting in hair growth [1]. The overexpression of TERT induces the proliferation of hair follicle stem cells (HFSCs) in the bulge region, leading to hair growth. In addition, Choi et al. (2008) showed that the presence of TERT in skin keratinocytes activates dormant HFSCs, which in turn activates hair follicles and promotes hair growth [2]. These studies suggest that TERT overexpression in dermal keratinocytes could be a target to promote hair growth. In a previous study, we constructed a system for screening food components that can activate the TERT promoter in keratinocytes and identified fisetin, a type of polyphenol. Fisetin activated keratinocytes, promoted hair growth, increased skin thickness, activated β-catenin, and increased the proliferation of HFSCs [3].

Recent studies have reported that exosomes derived from hair follicle tissues are involved in hair growth promotion and function as the molecular basis of cell–cell interactions [4,5,6,7,8,9]. In the present study, we aimed to determine whether fisetin activates keratinocytes, resulting in the activation of keratinocyte–HFSC interaction, which leads to hair growth. In addition, we sought to confirm the presence of exosomes as the molecular basis of this interaction.

## 2. Materials and Methods

### 2.1. Cell Line and Reagent

HaCaT cell line (human keratinocyte), obtained from Riken Bioresource Center (Tsukuba, Japan), was cultured in Dulbecco’s modified Eagle’s medium (Nissui, Tokyo, Japan) supplemented with 10% fetal bovine serum (Life Technologies, Gaithersburg, MD, USA) at 37 °C in a 5% CO_2_ atmosphere. Human HFSCs were obtained from Celprogen (Torrance, CA, USA) and cultured in human hair follicle stem cell complete growth medium with serum (Celprogen) at 37 °C in a 5% CO_2_ atmosphere. Fisetin (>96.0% purity; Tokyo Chemical Industry Co., Ltd., Tokyo, Japan) was purchased from

### 2.2. Quantitative Reverse Transcriptase-Polymerase Chain Reaction

RNA was extracted from cells and skin samples using the High Pure RNA Isolation Kit (Roche Diagnostics GmbH, Mannheim, Germany) and RNeasy Fibrous Tissue Mini Kit (Qiagen, Hilden, Germany), respectively, according to the manufacturers’ instructions. cDNA synthesis and quantitative reverse transcription-polymerase chain reaction (qRT-PCR) were performed, as described previously [3,10]. Samples were analyzed in triplicate, and gene expression levels were normalized to the corresponding β-actin gene level. The PCR primer sequences were described previously [3].

### 2.3. Investigation of Hair Growth in Experimental Animals

Six-week-old female C57BL/6 mice were obtained from Clea Japan (Tokyo, Japan) and acclimated with food and water ad libitum for one week. Nine mice were used for control and fisetin treatment, respectively. At 7 weeks of age, the dorsal hair of the mice was trimmed using electric hair clippers (Thrive Model 2100, Daito Electric Machine Industry, Osaka, Japan), shaved using an electric shaver (Panasonic, Osaka, Japan), and removed using a hair removal cream (CBS, Reckitt Benckiser, Tokyo, Japan), and all the hair follicles were synchronized in the telogen stage, as described previously [11,12]. This was followed by the daily topical application of 0.05 mL of 50% ethanol containing fisetin (1% *w/v*) using a spatula for 12 days, and hair growth was evaluated. Animal experiments were conducted in accordance with the “Guide for the Care and Use of Laboratory Animals” and approved by the Ethics Committee on Animal Experimentation of Kyushu University (approval number: A28-187-0).

### 2.4. Immunohistochemistry

Skin tissue sample preparations and treatments were described previously [3]. Paraffin-embedded hair follicles were cut out to a thickness of 5 micrometers. Immunohistochemical analysis was performed as previously described [3]. The tissues were first stained with primary antibodies (anti-Ki67, #12202, Cell Signaling Technology, Danvers, MA, USA; anti-β-catenin, #8480, Cell Signaling Technology; and anti-CD34, sc-74499, Santa Cruz Biotechnology, Santa Cruz, CA, USA) and then with secondary antibodies (Alexa Fluor 555 anti-rabbit IgG or Alexa Fluor 488 anti-mouse IgG, Thermo Fisher Scientific KK, Tokyo, Japan). After staining with Vectashield mounting medium (Vector Laboratories, Burlingame, CA, USA), the tissue samples were observed with the EVOS M5000 Imaging System (Thermo Fisher Scientific).

### 2.5. Evaluation of Mitochondrial Characteristics

Cells were stained with 250 nM of MitoTracker Red CMXRos (Thermo Fisher Scientific) and incubated at 37 °C for 30 min. Next, these cells were stained with 200 nM MitoTracker Green FM (Thermo Fisher Scientific) and incubated at 37 °C for 30 min. Finally, the cells were stained with Hoechst 33342 (Dojindo, Kumamoto, Japan), followed by incubation at 37 °C for 30 min. Then, the stained cells were analyzed using the IN Cell Analyzer 2200 (GE Healthcare, Amersham Place, UK), and the number, area, and activity of mitochondria were quantitatively determined using the IN Cell Investigator high-content image analysis software (GE Healthcare).

### 2.6. mRNA Microarray Assay

Total RNA was extracted from HaCaT cells treated with fisetin using Isogen II (Nippon Gene, Tokyo, Japan). Microarray analysis was performed, as described previously [10].

We identified genes with altered expression, as previously described [13]. Then, we established the following criteria for significantly upregulated or downregulated genes: upregulated genes, Z-score ≥ 2.0 and ratio ≥ 1.5-fold; downregulated genes, Z-score ≤ −2.0 and ratio ≤ 0.66-fold. To determine significantly over-represented gene ontology (GO) categories and significantly enriched pathways, we used the tools and data provided by the Database for Annotation, Visualization, and Integrated Discovery (http://david.abcc.ncifcrf.gov, 2 February 2021) [14,15].

### 2.7. Exosome Isolation

The MagCapure Exosome Isolation Kit PS (Fujifilm Wako Pure Chemical Corp., Osaka, Japan) was used to isolate exosomes from the medium of HaCaT cells treated with 10 μM fisetin, according to the manufacturer’s instructions [10,16].

### 2.8. Immunocytochemistry

The cells were fixed in 4% paraformaldehyde at 25 °C for 15 min. After washing with phosphate-buffered saline (PBS), the cells were blocked with blocking buffer (5% goat serum and 0.3% Triton X-100 in PBS) at room temperature for 1 h. After removing the buffer, the cells were incubated with primary antibodies (anti-Ki67, #12202; anti-active-β-catenin, #05-665, Merck Millipore, Billerica, MA, USA; and anti-TOMM20, ab186735, Abcam, Cambridge, UK) at 4 °C overnight. After washing with PBS, the cells were incubated with secondary antibodies (Alexa Fluor 555 anti-rabbit IgG, Alexa Fluor 488 anti-rabbit IgG, or Alexa Fluor 555 anti-mouse IgG) at 25 °C for 2 h. Then, the cells were washed and stained with Hoechst 33342 at room temperature for 20 min and observed under a fluorescence microscope (BZ-X800, Keyence, Osaka, Japan) [3].

### 2.9. Statistical Analysis

All experiments were performed at least three times, and the representative data are presented. The results are shown as the mean ± standard deviation. Statistical significance was determined using a two-sided Student’s *t*-test. Statistical significance was defined as *p* < 0.05 (* *p* < 0.05; ** *p* < 0.01; *** *p* < 0.001).

## 3. Results

### 3.1. Fisetin Augments the Expression of TERT in Keratinocytes

In our previous study, we observed that fisetin augmented the expression of TERT in a human keratinocyte cell line and the dorsal skin cells of mice [3]. In the present study, we evaluated the expression of TERT in dorsal skin cells of mice treated with fisetin. The results clearly showed that fisetin augmented TERT expression in the dorsal skin cells of mice (Figure 1A).

### 3.2. Fisetin Promotes Hair Growth in Mice

Since the functions and properties of the skin change significantly with the hair cycle, it is generally considered important to control the hair cycle in animal experiments on hair growth. To obtain a uniform hair cycle, it is widely known that hair removal creams and electric shavers can be used to simultaneously remove hair within a certain range of the telogen phase and stimulate them to induce a uniform anagen phase [12]. Hair growth was strongly promoted in mice in the fisetin group compared to those in the control group (Figure 1B). The degree of hair growth was scored using skin color and hair growth as indicators, and the results showed that hair growth was significantly enhanced in the fisetin group compared to the control group (Figure 1C).

### 3.3. Fisetin Activates HFSCs In Vivo

After the hair growth test was completed, the condition of the HFSCs was observed using paraffin-embedded sections of the skin using antibodies against β-catenin, CD34 (a stem cell marker), and Ki67 (a cell proliferation marker). The results showed that β-catenin and Ki67 were strongly expressed in CD34^+^ cells in the vicinity of hair follicles on the dorsal side of the skin of fisetin-treated mice (Figure 2). This suggested that the proliferation of HFSCs in the vicinity of the dorsal skin of fisetin-treated mice was activated and that fisetin activated the interaction between keratinocytes and HFSCs, resulting in the promotion of hair growth.

### 3.4. Fisetin Activates HaCaT Cells

First, we examined the effects of fisetin on keratinocytes. The results showed that fisetin augmented the expression of SIRT1 and further increased and activated the mitochondria in HaCaT cells (Figure 3A,B). In addition, fisetin regulated the expression of various genes, such as those encoding secretory factors and those involved in epidermogenesis, cell proliferation, hair follicle regulation, and hair cycle regulation (Table 1 and Table 2). These results suggest that fisetin activates HaCaT cells.

### 3.5. Effects of Exosomes Derived from Fisetin-Treated HaCaT Cells on HFSCs

In this study, we focused on exosomes as the mediators of the interaction between keratinocytes and HFSCs and hypothesized that the exosomes secreted by keratinocytes activated by fisetin would activate the HFSCs. Therefore, we first prepared exosomes from the supernatant of fisetin-treated HaCaT cells and tested their effect on the proliferation of HFSCs. The results showed that fisetin-treated HaCaT cell-derived exosomes triggered the nuclear translocation of β-catenin (Figure 4A), augmented the expression of AXIN2 (Figure 4B), and increased the number of Ki67^+^ cells in HFSCs (Figure 4C). These results suggest that fisetin-treated HaCaT cell-derived exosomes activate the proliferation of HFSCs.

We next tested whether fisetin-treated HaCaT cell-derived exosomes could activate the mitochondria in HFSCs. The HFSCs were treated with fisetin-treated HaCaT cell-derived exosomes, and the expression of TOMM20, a mitochondrial marker, was verified by immunostaining. The results showed that the number of TOMM20^+^ cells was significantly increased in the HFSCs (Figure 5A). Moreover, the number and activity of mitochondria were evaluated using the fluorescence probes MitoTracker Green FM and MitoTracker Red CMXRos, respectively. The results showed that fisetin-treated HaCaT cell-derived exosomes significantly increased the number of cells harboring activated mitochondria relative to the total number of mitochondria (Figure 5B).

## 4. Discussion

In a previous study, we reported that fisetin, which augments the expression of TERT in keratinocytes, induces a shift from telogen to anagen in the hair follicles by inducing the proliferation of HFSCs, thus promoting hair growth [3]. These results suggest that fisetin activates keratinocytes and further strengthens the interaction between keratinocytes and HFSCs. In the present study, we aimed to clarify the molecular basis of fisetin-induced hair growth.

First, after confirming that fisetin is not toxic to keratinocyte at concentrations up to 10 μM, we examined whether keratinocytes are activated by fisetin. We found that fisetin activated the SIRT1-mitochondrial axis as well as keratinocytes by regulating the expression of various genes, such as those encoding secretory factors and those involved in epidermogenesis, cell proliferation, hair follicle regulation, and hair cycle regulation. Since mitochondrial dysfunction in skin cells is believed to cause an increase in reactive oxygen species levels and may be a cause of skin aging [17], activation of the SIRT1–mitochondrial axis by fisetin may lead to keratinocyte improvement and activation [18]. These results suggest that fisetin activates keratinocytes, which could lead to the promotion of hair growth.

Next, we examined the molecular basis for the activation of the keratinocyte–HFSC interaction induced by fisetin. We assumed that exosomes are involved in the activation of cell–cell interactions. Exosomes are an important component of paracrine signaling and can mediate communication between distant cells by directly transferring various biomolecules, including miRNAs, from donor cells to recipient cells [10,19,20]. However, the potential of exosomes as modulators of hair follicle dynamics has not received widespread attention. Previous studies have shown that exosomes derived from dermal papillae [4,5], dermal fibroblasts [6], and mesenchymal stem cells [7,8,9] can activate the proliferation of HFSCs and consequently induce hair growth. Key findings regarding the relevance of exosomes in skin and hair follicle regeneration have been reported [21], with an emphasis on the signaling pathways that mediate these effects. It is widely known that Wnt is a master regulator of hair follicle morphogenesis and hair growth [22]. Furthermore, mitochondrial aerobic respiration is activated during HFSC differentiation [23]. Thus, activation of the Wnt pathway and mitochondria is essential for hair follicle activation and differentiation.

In the present study, we found that fisetin-treated HaCaT cell-derived exosomes activated β-catenin and promoted HFSC growth, in addition to activating the mitochondria. These results revealed exosomes as the molecular basis of keratinocyte–HFSC interaction and that fisetin, along with its effects on keratinocytes, could cause exosome secretion and the consequent activation of HFSCs. Furthermore, since hair growth is triggered by exosome secretion, the hair growth-promoting effect is expected to continue for some time after fisetin application is discontinued. To the best of our knowledge, this is the first study to show that keratinocyte-derived exosomes can activate HFSCs, which in turn induces hair growth.

In conclusion, based on the obtained results, exosomes can be considered as natural mediators that could be involved in hair cycle regulation as well as serve as promising delivery vehicles for improving skin and hair regeneration because of their potential to target various molecular processes and cells. Further studies are required to clarify the mode of involvement of exosomes in the hair cycle in vivo and the therapeutic effects that can be expected when using exosomes to regulate hair growth in clinical settings.

## Figures and Tables

**Figure 1 nutrients-13-02087-f001:**
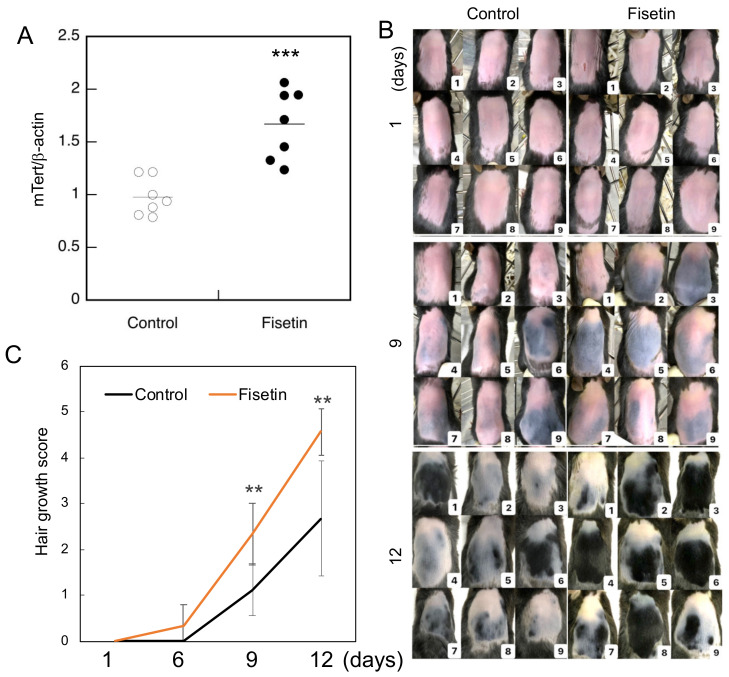
Hair growth-promoting effect of fisetin. (**A**) The expression of *mTert* in the dorsal skin cells of fisetin-treated mice and investigated using qRT-PCR. (**B**) Evaluation of the effect of fisetin treatment on the dorsal skin of C57BL/6 mice and hair growth after 12 days. (**C**) Hair growth score of fisetin-treated and control groups (the entire shaved back is pink = 0; part of the shaved back is blue = 1; part of the shaved back is gray = 2; the entire shaved back is gray = 3; part of the shaved back is black = 4; the entire shaved back is black = 5). Statistical significance was determined using a two-sided Student’s t-test. Statistical significance was defined as *p* < 0.05 (** *p* < 0.01; *** *p* < 0.001).

**Figure 2 nutrients-13-02087-f002:**
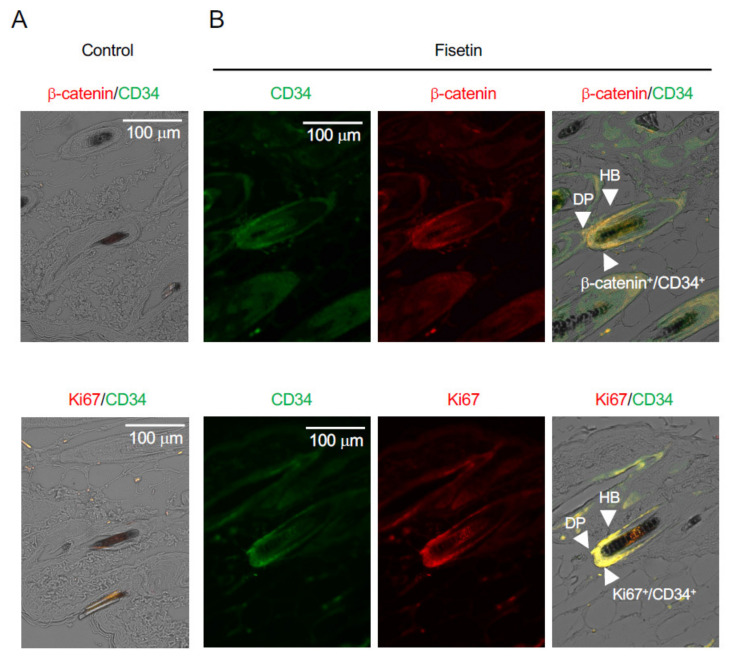
Effects of fisetin treatment on the expression of marker proteins in dorsal skin sections. Immunohistochemistry analysis of skin sections was performed using anti-β-catenin, anti-Ki67, and anti-CD34 antibodies ((**A**), Control; (**B**), Fisetin) (DP, dermal papilla; HB, hair bulge).

**Figure 3 nutrients-13-02087-f003:**
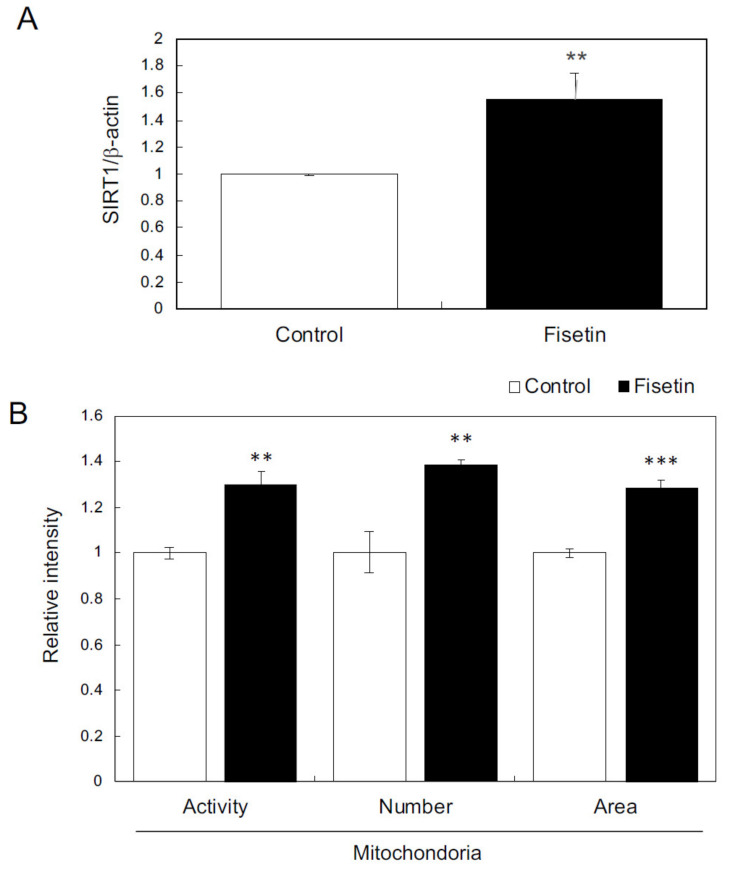
Effects of fisetin on keratinocytes. (**A**) The expression of *SIRT1* in HaCaT cells was evaluated using qRT-PCR. (**B**) Effects of fisetin on mitochondria evaluated using mitochondrion-specific probes (MitoTracker Red CMXRos and MitoTracker Green FM) and IN Cell Analyzer 2200. Statistical significance was determined using a two-sided Student’s t-test. Statistical significance was defined as *p* < 0.05 (** *p* < 0.01; *** *p* < 0.001).

**Figure 4 nutrients-13-02087-f004:**
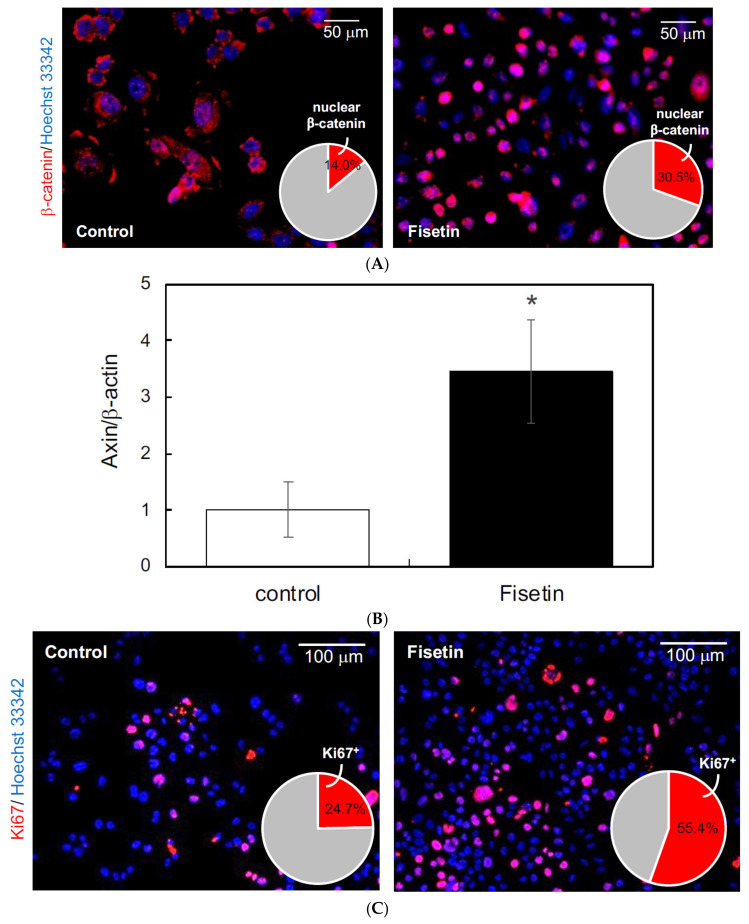
Effects of exosomes derived from fisetin-treated keratinocytes on hair follicle stem cells (HFSCs). (**A**) Localization of β-catenin in HFSCs was evaluated by immunocytochemistry using anti-β-catenin antibody. Nuclear localization of β-catenin (red in pie chart) in HFSCs treated with exosomes was quantitatively determined using IN Cell Analyzer 2200. (**B**) Expression of axin in HFSCs treated with exosomes was quantitatively determined using qRT-PCR. (**C**) Ki67^+^ cells in HFSCs were detected by immunocytochemistry using anti-Ki67 antibody. The number of Ki67^+^ cells (red in pie chart) was determined using IN Cell Analyzer 2200. Statistical significance was determined using a two-sided Student’s t-test. Statistical significance was defined as *p* < 0.05 (* *p* < 0.05).

**Figure 5 nutrients-13-02087-f005:**
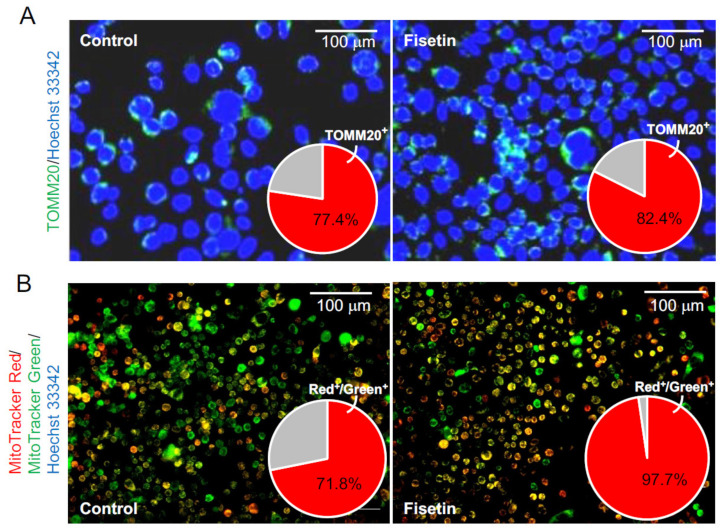
Effects of exosomes derived from fisetin-treated keratinocytes on the mitochondria in HFSCs. (**A**) TOMM20^+^ cells in HFSCs were detected by immunocytochemistry using anti-TOMM antibody. The number of TOMM20^+^ cells (red in pie chart) was determined using IN Cell Analyzer 2200. (**B**) Mitochondria in HFSCs were detected by immunocytochemistry using specific probes (MitoTracker Red CMXRos and MitoTracker Green FM). The relative number of cells harboring active mitochondria (yellow cells in photo; red in pie chart) was determined using IN Cell Analyzer 2200.

**Table 1 nutrients-13-02087-t001:** Functional annotation.

Annotation Cluster	Term	Count	*p*-Value
1	Oxidoreductase	50	4.90 × 10^−6^
	Oxidation-reduction process	49	9.90 × 10^−5^
2	Secreted	122	2.20 × 10^−5^
	Signal	214	1.60 × 10^−3^
3	Palmoplantar keratoderma	10	5.20 × 10^−6^
	Keratin	10	2.10 × 10^−1^
4	Regulation of cell growth	11	3.60 × 10^−3^
	Insulin-like growth factor-binding	4	4.60 × 10^−2^
	Negative regulation of cell death	4	4.70 × 10^−1^

**Table 2 nutrients-13-02087-t002:** GO analysis.

GO Term	Genes
Hair follicle regulation system gene	INHBA, RUNX1, TGFB2, FST, KRT17
Hair cycle control gene	KRT14

## Data Availability

The data presented in this study are openly available in QIR at [https://catalog.lib.kyushu-u.ac.jp/opac_browse/papers/?lang=0, accessed on 18 June 2021].

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
