# Peer review of "Exosomes Derived from Fisetin-Treated Keratinocytes Mediate Hair Growth Promotion"

_nutrients, 2021, doi:10.3390/nu13062087_

Round 1

Reviewer 1 Report

Could you explain why mice 1 in the fisetin group seemed to have less growth than many of the mice in the control group?

The Figure 2 that shows immunohistochemical expresion, what part of the follicle is showing? Please state in the legend

Why there is absence of CD34 positive cells in the skin of the control mice? normally CD34 positive cells should be seen in the permanent bulge portion of the follicle at all stages of the hair cycle.

Author Response

Reviewer #1:

Thank you for your valuable comments. I revised the manuscript according to the reviewer’s comments.

Comment #1:

Could you explain why mice 1 in the fisetin group seemed to have less growth than many of the mice in the control group?

Response #1:

I don't know the exact reason, but it may be that the fisetin was not applied correctly to the mouse 1.

Comment #2:

The Figure 2 that shows immunohistochemical expression, what part of the follicle is showing? Please state in the legend.

Response #2:

According to the comment, we revised the Figure 2 and its legend.

Comment #3:

Why there is absence of CD34 positive cells in the skin of the control mice? normally CD34 positive cells should be seen in the permanent bulge portion of the follicle at all stages of the hair cycle.

Response #3:

As the reviewer pointed out, CD34 positive cells are indeed present in the bulge region of the hair follicle of control mice. But the number is very small, and Fig. 2 shows that the fisetin treatment increased the number.

Reviewer 2 Report

The Paper by Ogawa et al., I.D. nutrients-1244864, submitted to Nutrients for publication deals with clarify the biological mechanism of HSFCs activation by keratinocyte-derived exosomes treated with fisetin. In my opinion, the work is well structured and described. The results are noteworthy.

  1. Please clarify in M&M where and with what degree of purity you bought fisetin. In case it was obtained by extraction, I ask you to indicate the procedure used for the isolation.
  2. In the paragraph 2.3, line 72: please clarify if the fisetin concentration (1%) is w/w or w/vol, and the solvent employed (50% ethanol, ok, and the rest?)
  3. Please, specify how many animals were in the control group and how many in the treated group in M&M section, clarifying which treatment the control group underwent: no application or only vehicle?
  4. Have the authors possibly assessed what happens to HFSCs after cessation of fisetin treatment? Does the activation of the Keratinocytes and the release of the exosomes stop or persist?
  5. Fisetin is known to induce DNA damage in some cell models (i.e. RPE cells, The Journal of Nutritional Biochemistry, 42, 2017, 37-42). Have the AA assessed the possible toxicity profile for medium / long-term exposure of the keratinocytes?

In my opinion it would be useful for the reader if the Authors express their considerations regarding these last two points in the Discussion paragraph.

Author Response

Reviewer #2:

Thank you for your valuable comments. I revised the manuscript according to the reviewer’s comments.

Comment #1:

Please clarify in M&M where and with what degree of purity you bought fisetin. In case it was obtained by extraction, I ask you to indicate the procedure used for the isolation.

Response #1:

We purchased fisetin (>96.0% purity) from Tokyo Chemical Industory Co., Ltd., and revised the manuscript accordingly.

Comment #2:

In the paragraph 2.3, line 72: please clarify if the fisetin concentration (1%) is w/w or w/vol, and the solvent employed (50% ethanol, ok, and the rest?).

Response #2:

According to the comment, we revised the manuscript.

Comment #3:

Please, specify how many animals were in the control group and how many in the treated group in M&M section, clarifying which treatment the control group underwent: no application or only vehicle?

Response #3:

According to the comment, we revised the manuscript.

Comment #4:

Have the authors possibly assessed what happens to HFSCs after cessation of fisetin treatment? Does the activation of the Keratinocytes and the release of the exosomes stop or persist?

Response #4:

Unfortunately, we have not verified this point, but since the fisetin effect is accompanied by exosome secretion, we believe that the effect will persist for some time after fisetin application is discontinued. According to the comment, we revised the discussion section.

Comment #5:

Fisetin is known to induce DNA damage in some cell models (i.e. RPE cells, The Journal of Nutritional Biochemistry, 42, 2017, 37-42). Have the AA assessed the possible toxicity profile for medium / long-term exposure of the keratinocytes?

Response #5:

We have confirmed that fisetin is not toxic to keratinocyte at concentrations up to 10 μM. According to the comment, we revised the discussion section.